# Structure determination of the HgcAB complex using metagenome sequence data: insights into microbial mercury methylation

Connor J. Cooper [1,2], Kaiyuan Zheng [3], Katherine W. Rush [3], Alexander Johs [4], Brian C. Sanders [2], Georgios A. Pavlopoulos [5,6], Nikos C. Kyrpides [5,7], Mircea Podar [1,2], Sergey Ovchinnikov [8], Stephen W. Ragsdale [3] & Jerry M. Parks [1,2✉]

Bacteria and archaea possessing the *hgcAB* gene pair methylate inorganic mercury (Hg) to form highly toxic methylmercury. HgcA consists of a corrinoid binding domain and a transmembrane domain, and HgcB is a dicluster ferredoxin. However, their detailed structure and function have not been thoroughly characterized. We modeled the HgcAB complex by combining metagenome sequence data mining, coevolution analysis, and Rosetta structure calculations. In addition, we overexpressed HgcA and HgcB in *Escherichia coli,* confirmed spectroscopically that they bind cobalamin and [4Fe-4S] clusters, respectively, and incorporated these cofactors into the structural model. Surprisingly, the two domains of HgcA do not interact with each other, but HgcB forms extensive contacts with both domains. The model suggests that conserved cysteines in HgcB are involved in shuttling $Hg^{II}$, methylmercury, or both. These findings refine our understanding of the mechanism of Hg methylation and expand the known repertoire of corrinoid methyltransferases in nature.

[1] Graduate School of Genome Science and Technology, University of Tennessee, F225 Walters Life Science, Knoxville, TN 37996, USA. [2] Biosciences Division, Oak Ridge National Laboratory, 1 Bethel Valley Road, Oak Ridge, TN 37831-6038, USA. [3] Department of Biological Chemistry, University of Michigan Medical School, 1150 West Medical Center Drive, Ann Arbor, MI 48109-0606, USA. [4] Environmental Sciences Division, Oak Ridge National Laboratory, 1 Bethel Valley Road, Oak Ridge, TN 37831-6038, USA. [5] DOE Joint Genome Institute, Lawrence Berkeley National Laboratory, 1 Cyclotron Road, Berkeley, CA 94720, USA. [6] Institute for Fundamental Biomedical Research, Biomedical Science Research Center "Alexander Fleming", 34 Fleming Street, 16672 Vari, Greece. [7] Environmental Genomics and Systems Biology Division, Lawrence Berkeley National Laboratory Berkeley, California, USA. [8] John Harvard Distinguished Science Fellowship Program, Harvard University, Cambridge, MA 02138, USA. ✉email: parksjm@ornl.gov

Anaerobic bacteria and archaea possessing the *hgcAB* gene pair methylate inorganic mercury (Hg) to form methylmercury (CH₃Hg⁺)[1–4], a potent neurotoxin. Deletion of *hgcA*, *hgcB*, or both completely abolishes the ability of microorganisms to make methylmercury. These genes are distributed somewhat sporadically among various Proteobacteria (*Deltaproteobacteria*), Firmicutes, and Euryarchaeota. They are also found in some Chloroflexi (*Dehalococcoides*), Chrysiogenetes, Nitrospirae, and others.

The *hgcAB* gene pair is relatively rare, occurring in only ~1.4% of sequenced microbial genomes[5]. Nevertheless, microorganisms harboring these genes are distributed worldwide in highly diverse anaerobic settings, including soils, sediments, periphyton, rice paddies, invertebrate digestive tracts, and various extreme environments. It is not known why microorganisms methylate Hg, but this process is generally not thought to be a Hg detoxification mechanism because microorganisms harboring *hgcAB* genes are apparently no less susceptible to Hg toxicity than those lacking them[6].

Protein sequence analysis revealed that HgcA (a subset of the CO dehydrogenase/acetyl-CoA synthase delta subunit family, PF03599) is a corrinoid (i.e., vitamin B₁₂-dependent) protein consisting of an N-terminal corrinoid binding domain (CBD) and a C-terminal transmembrane domain (TMD) with five TM helices[1]. The CBD of HgcA bears homology to the C-terminal domain of the large subunit of the corrinoid iron–sulfur protein (CFeSP) from the Wood-Ljungdahl pathway in acetogenic bacteria[7–9].

HgcA was predicted to include a "cap helix" in its CBD similar to that in CFeSP[7]. The cap helix in CFeSP interacts noncovalently with the α face of the corrinoid cofactor. In HgcA, the putative cap helix region includes several highly conserved residues, one of which is a strictly conserved Cys residue (Cys93 in *Desulfovibrio desulfuricans* ND132), that is not present at the corresponding position in the sequence of CFeSP. On the basis of its position in a homology model of the CBD, this Cys residue was predicted to bind the corrinoid cofactor in a cobalt-thiolate, or "Cys-on" configuration[1]. Findings from in vivo site-directed mutagenesis experiments are consistent with Cys-on cofactor binding[10]. Mutation of Cys93 to Ala or Thr resulted in a complete loss of Hg methylation activity, but a His mutant, which can presumably still coordinate with Co, retained partial activity. In addition, substitution of several amino acids in the cap helix region with a helix-breaking Pro residue drastically reduced or completely abolished activity. A quantum chemical study showed that Cys-on coordination promotes the exchange of one organometallic (Co–C) bond for another (Hg–C)[11]. Recently, the first example of Cys-on coordination in a protein was observed for the bacterial vitamin B₁₂ transporter BtuM co-crystallized with cobalamin[12].

The TMD of HgcA has no detectable sequence homology (i.e., BLAST *E*-value < 10) to any structurally characterized protein. C-terminal truncation mutants of HgcA in which the TMD was deleted by introducing a stop codon after the nucleotides encoding either amino acid 166 or 187 were both unable to methylate Hg, indicating that this domain is essential for activity[10].

HgcB is a 10.2 kDa bacterial ferredoxin (Pfam entries PF13237 and PF00037) that includes two CxxCxxCxxxCP motifs, which are known to bind [4Fe-4S] clusters. In addition, HgcB includes another strictly conserved Cys (Cys73 in *D. desulfuricans* ND132), located ~12 residues downstream of the second [4Fe-4S]-binding motif, and up to four additional Cys residues at its C-terminus. Two cysteines are present at the C-terminus of ND132 (Cys94 and Cys95). Homologs of HgcB have variable sequence length, in particular in the tail region near the C-terminus. Mutation of Cys73 to Ala completely abolished Hg methylation

in vivo[4]. Mutation of either C-terminal cysteine (Cys94 or Cys95) individually to Ala did not affect Hg methylation activity, but mutation of both residues simultaneously to Ala led to a 95% reduction in activity compared to the wild-type. Thus, at least one Cys is required at the C-terminus for maximal Hg methylation activity.

In a proteomics study of *Geobacter sulfurreducens* PCA, another confirmed Hg-methylating bacterium, HgcA and HgcB were not detected due to low-protein abundance[13]. In a subsequent study of *D. desulfuricans* ND132, HgcA was detected in low abundance but HgcB was again not detected[14]. Thus, isolation and purification of sufficient quantities of protein from a native host are expected to be challenging. Heterologous over-expression of HgcA and HgcB is complicated by a number of factors. For example, many Hg-methylating organisms are obligate anaerobes. Based on the proposed Hg methylation cycle, maintaining a low redox potential is essential for the function of HgcA and HgcB. It has been demonstrated that exposure to oxygen inhibits MeHg formation in cell lysates of *D. desulfuricans* ND132[15]. In addition, incorporation of the corrinoid cofactor and [4Fe-4S] clusters is nontrivial in heterologous hosts such as *Escherichia coli* because the uptake of corrinoids is tightly regulated[16] and overexpression of recombinant proteins increases the demand on the machinery required to assemble iron–sulfur clusters[17]. Lastly, although tremendous progress has been made in recent years, structure determination of transmembrane proteins with X-ray crystallography, nuclear magnetic resonance, or cryo-electron microscopy remains a challenge.

In the absence of an experimentally determined structure, structural modeling is a viable means for obtaining mechanistic insight into protein function. Homology modeling is generally the method of choice, provided that suitable template structures are available. When templates are lacking, however, models can be generated by leveraging coevolution information inferred from a multiple sequence alignment. Pairs of amino acids that coevolve are likely to be in close spatial proximity in the folded protein. Thus, by imposing contact restraints derived from coevolution analysis with ab initio protein modeling, accurate structural models can be obtained[18–22].

Coevolution analysis requires as input a multiple sequence alignment with a large number of sequences. The massive amount of data available in public repositories such as the UniRef100 database[23] and the DOE Joint Genome Institute (JGI) metagenome database[24] provide a rich source of diverse protein sequences. Recently, it was shown that the combination of metagenome sequences, coevolution analysis and Rosetta protein structure calculations can produce highly accurate structures[25]. For a multiple sequence alignment, when the effective number of sequences divided by the square root of the sequence length *L* is >64 (where the effective number of sequences is defined as 1 over the number of sequences within 80% identity), then homology model-level accuracy or better can be obtained.

Structural models of HgcA and HgcB would provide valuable insight into the biochemical mechanism of Hg methylation. Here, we express HgcA and HgcB individually in *E. coli* and show by UV–visible spectroscopy that they indeed bind corrinoid and iron–sulfur cofactors, as predicted from previous bioinformatics analyses. We then combine metagenome-based protein structure calculations to generate models of the individual domains of HgcA and of HgcB. We then show how these domains assemble to form the HgcAB complex and incorporate a vitamin B₁₂ corrinoid cofactor and two [4Fe-4S] clusters into the model. In addition, we analyze >4300 genomic and metagenomic sequences of HgcA to show that the evolution of this enzyme family has been marked by extensive horizontal gene transfer. A large

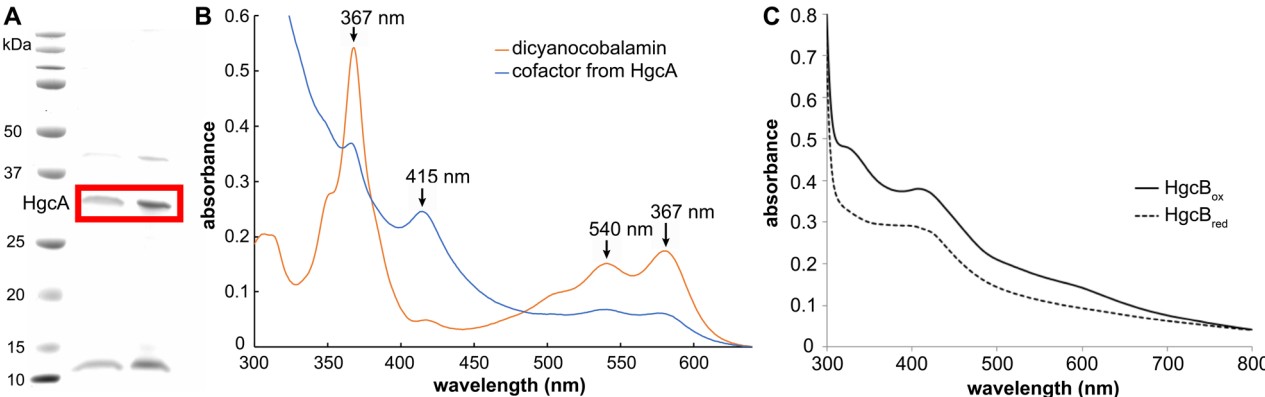

**Fig. 1 Protein purification and spectral characterization. a** SDS-PAGE gel of purified HgcA. The bands enclosed in the red rectangle are HgcA in elution buffer and after buffer exchange, respectively, as verified by western blot analysis using an antibody against the His-tag. A full, uncropped gel image is provided in Supplementary Fig. 1. **b** UV–visible spectrum of dicyanocobalamin (orange) and cofactor extracted from purified, His-tagged HgcA by heating to 95 °C with KCN (blue). HgcA was dissolved in phosphate buffer (50 mM $K_2HPO_4$, 100 mM NaCl, 10% glycerol, 2 mM BME, 10 mM imidazole, pH 7.4). **c** UV–visible spectrum of oxidized, as-isolated MBP-HgcB ($HgcB_{ox}$) and MBP-HgcB after reduction with sodium dithionite ($HgcB_{red}$).

diversity of HgcA is present in organisms that have not yet been cultured.

## Results

We cloned and expressed full-length HgcA from *D. desulfuricans* ND132 heterologously in *E. coli* as an N-terminal His-tagged construct (His-HgcA) (Fig. 1a). Similarly, HgcB was produced separately as a maltose-binding protein fusion construct (MBP-HgcB).

**Electronic spectra of HgcA and HgcB**. After purifying each protein, we obtained UV–visible spectra to confirm cofactor binding. The characteristic UV–visible peaks of dicyanocobalamin are 367, 540, and 580 nm[26,27]. We obtained a spectrum from KCN and heat-treated His-HgcA (95 °C for 20 min) and compared it to that of of 20 µM dicyanocobalamin dissolved in the same phosphate buffer (Fig. 1b). Both spectra show the characteristic peaks of dicyanocobalamin, demonstrating that HgcA indeed binds cobalamin. Sodium dithionite (1 mM) was added to 12.5 µM HgcB (25 µM [4Fe-4S] cluster), quenching the absorbance in the 300–500 nm region, as is characteristic of reduced [4Fe-4S] cluster proteins (Fig. 1c). UV–visible data are provided in Supplementary Data 1.

**Lack of suitable templates for homology modeling**. Structural models of HgcA published to date are limited to the core of the CBD[1,28]. To determine whether including coevolutionary information is likely to provide more information for structural modeling of HgcA and HgcB than homology modeling, we searched a nonredundant subset of structures in the Protein Data Bank and calculated HHΔ for potential templates (see Methods). HHΔ values <0.5 for a query and template sequence are generally considered to be good candidates for template-based modeling, whereas those with values >0.5 are not. The lowest HHΔ value for the paired alignment of HgcA and HgcB is 0.77 (Supplementary Table 1), with the top hit corresponding to an X-ray structure of the corrinoid iron–sulfur protein CFeSP (PDB entry 4DJD)[8]. However, only the core of the CBD is covered by the template. No structures in the PDB were identified by *hhsearch* that could serve as templates for the TMD of HgcA. The lowest HHΔ value for a template that covers HgcB is 0.92 for the Fe hydrogenase from *D. desulfuricans* (PDB entry 1HFE)[29].

**Multiple sequence alignments and contact map predictions**. To obtain a sufficient number of sequences for coevolution analysis, we searched a large master database comprising JGI metagenomes and the UniRef100 database for sequence homologs of HgcA and HgcB. Initial searches identified 7505 and 19,317 putative HgcA and HgcB sequences, respectively. We then exploited co-occurrence and adjacency to generate a paired alignment of HgcA and HgcB. After pairing of HgcA and HgcB sequences based on whether two hits were from the same metagenomic contig, we obtained 3025 sequences. We used 90% identity filtering to remove redundant sequences (2432), but later reweighted by 80% identity to obtain the effective number of sequences (1783). From the paired alignment, the estimated contact prediction accuracy is $N_f = \text{seq}/\sqrt{\text{len}} = 87.1$ for the 419 amino acids in HgcA and HgcB remaining after trimming regions at the N- and C-termini that are not well constrained by predicted contacts. This $N_f$ value indicates that HgcA and HgcB are excellent candidates for structural modeling guided by coevolution-based contact restraints.

**Structural modeling**. Intra- and interdomain residue-residue contacts were predicted by performing a coevolution analysis of the HgcAB paired alignment. Surprisingly, the contact map includes very few predicted contacts between the two domains of HgcA (Fig. 2). Gly33 is predicted to interact with Val186, and Leu32 is predicted to interact with Tyr189. In addition, Val173 and Thr174 are both predicted to interact with Glu179, but these residues are located near the boundary between the two domains. However, there is clear evidence for several contacts between the CBD of HgcA and HgcB.

**CBD of HgcA**. Rosetta modeling guided by coevolution analysis revealed that the core of the CBD of HgcA adopts a Rossmann fold with five β-sheets, four major α-helices and two short helical regions (Fig. 3). An additional α-helix is present near the N-terminus. A search of the Protein Data Bank with the Dali web server revealed several proteins with structural similarity to the CBD model (Supplementary Table 2). As expected, the protein with the greatest structural similarity to the CBD of HgcA is CFeSP (PDB entry 2YCL, Z-score = 14.2). The sequence identity between the CBD of HgcA (residues 15–166) and CFeSP (residues 291–445) is only 27%, but the binding pocket that accommodates the nucleotide tail of the cofactor is similar in the two proteins[1]. Besides the four conserved hydrogen bonds that were

used as distance restraints (see Methods), the $B_{12}$ cofactor forms hydrogen bonds with several other residues in the model (Fig. 3 and Supplementary Table 3).

**TMD of HgcA**. The TMD consists of five TM helices, with helix 4 forming a central stalk that is mostly surrounded by helices 1, 2, 3, and 5 (Fig. 4). Helices 1 and 2 are the longest, both consisting of 31 residues. Helix 5 includes 29 residues and helix 4 includes 24. Helix 3 is the shortest, comprising 21 residues. Based on the coevolution analysis, all adjacent pairs of helices in the model are predicted to be in contact with each other except for helices 1 and 5 (Fig. 4b). A search of the Protein Data Bank with the Dali web server identified structural similarity between the TMD of HgcA and several membrane proteins (Supplementary Table 4). Interestingly, the top hit is an X-ray structure of the homodimeric $Mg^{2+}$ transporter MgtE from *Thermus thermophilus* (PDB entry 2YVX, Z-score = 6.8)[30].

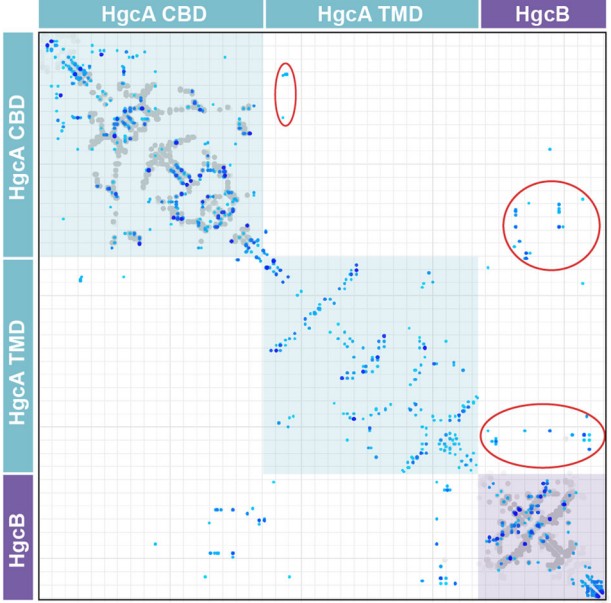

**Fig. 2 Contact map predicted from coevolution analysis of the paired HgcAB multiple sequence alignment.** Contacts are shown in shades of blue (darker blue = higher probability) and contacts from X-ray structures of homologs (Supplementary Table 1) are shown in gray. Individual domains are labeled and interdomain contacts are circled in red.

**HgcB**. HgcB consists of an N-terminal core domain with a typical [4Fe-4S] ferredoxin fold[31] followed by an α-helical extension and a disordered tail at its C-terminus (Fig. 5). The core domain of HgcB (residues 12–68) displays the same twofold pseudosymmetry as the bacterial ferredoxin from *Clostridium acidurici* (PDB entry 2FDN[32]) and other ferredoxins. In addition, it is structurally similar to numerous proteins including heterodisulfide reductase, tungsten formylmethanofuran dehydrogenase subunit FwdA, photosystem I subunit PsaC, and adenylylsulfate reductase (Supplementary Table 5). A similar α-helical extension is present in some ferredoxins, such as that from *Thauera aromatica*[33]. However, the additional disordered tail at its C-terminus appears to be unique to HgcB.

Cysteine residues 20, 23, 26, and 60 bind cluster A and residues 50, 53, 56, and 30 bind cluster B. The strictly conserved Cys73 in HgcB is located at the beginning of the α-helical extension and is located ~13 Å from the nearest Fe atom in cluster B in the model (Fig. 5b). The number of cysteines and the total number of residues in the disordered tail vary among HgcB orthologs (Supplementary Data 2). Of the 2432 sequences in the paired alignment, 1943 have at least one additional Cys located downstream of Cys73 and 1317 have two or more C-terminal cysteines. The majority of these sequences were obtained from metagenomes, so it is likely that some are truncated at their termini. Thus, these counts represent a lower bound for the number of cysteines located at or near the C-terminal tail of HgcB.

**Assembly and analysis of the HgcAB complex**. Using the top predicted interdomain contacts to guide docking of the individual domains together (Supplementary Fig. 2), we generated a model of the HgcAB complex (Fig. 6). Based on the ratio of the number of contacts in the model to those expected from the coevolution analysis given the number of sequences in the paired alignment and the GREMLIN score[22], the estimated accuracy of the model, $R_c$, is 0.87. $R_c$ values for native proteins range from 0.7 to 1.2. Thus, in general the HgcAB structural model fits the predicted contact set well (Supplementary Fig. 3).

**Interfacial residues**. In the assembled complex, residues in the CBD of HgcA interact with the core of HgcB via several polar contacts: Gly96 (O)–Arg58 (NE), Gly132 (N)–Asn59 (OD1), Thr131 (OG1)–Asn59 (OD1), Arg136 (NH1)–Pro61 (O), Gly132 (O)–Ser25 (OG), Glu168 (OE2)–Lys2 (NZ), and Val (N)–Pro31 (O) (Supplementary Fig. 4 and Supplementary Table 6). Polar contacts between residues in the TMD of HgcA and HgcB include: Asn245 (O)–Arg5 (NH1), Arg250 (NH2)–Arg5 (O),

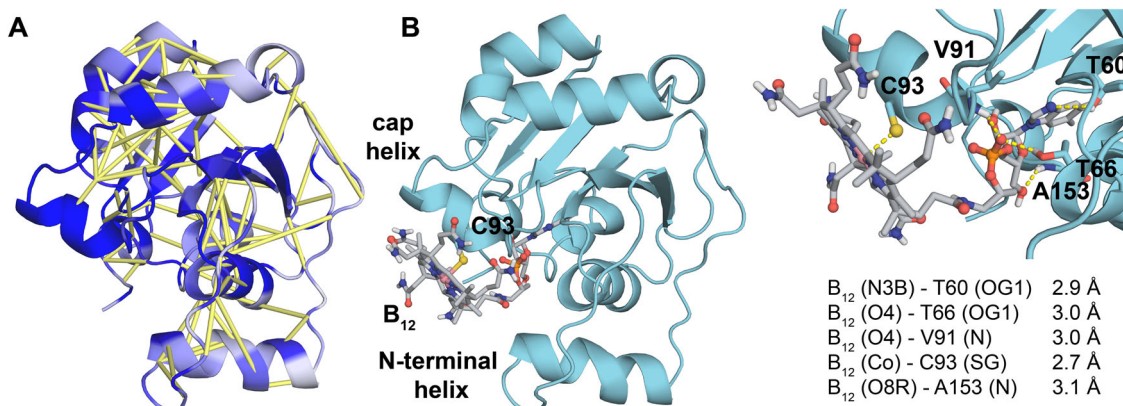

| | |
|---|---|
| $B_{12}$ (N3B) - T60 (OG1) | 2.9 Å |
| $B_{12}$ (O4) - T66 (OG1) | 3.0 Å |
| $B_{12}$ (O4) - V91 (N) | 3.0 Å |
| $B_{12}$ (Co) - C93 (SG) | 2.7 Å |
| $B_{12}$ (O8R) - A153 (N) | 3.1 Å |

**Fig. 3 Model of the corrinoid binding domain of HgcA. a** Predicted contacts are shown as yellow bars and residues colored according to sequence conservation (dark blue = highest). **b** Model including the $B_{12}$ cofactor. Key residues and distances are shown.

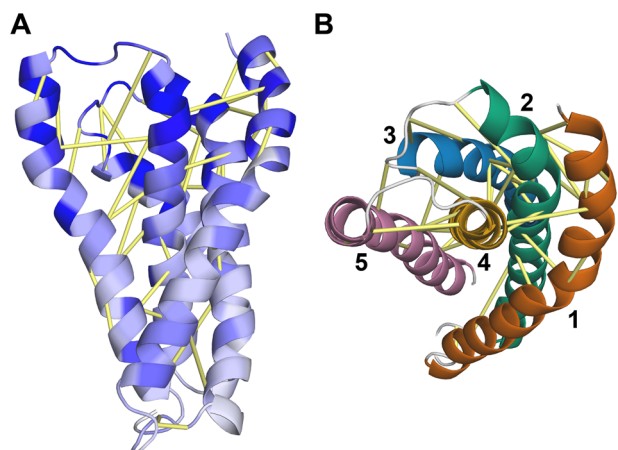

**Fig. 4 Model of the transmembrane domain of HgcA. a** Predicted contacts are shown as yellow bars and residues are colored according to sequence conservation (dark blue = highest). **b** Model rotated ~90 degrees forward to show a "top" view from the cytoplasmic side. Each helix is shown in a different color.

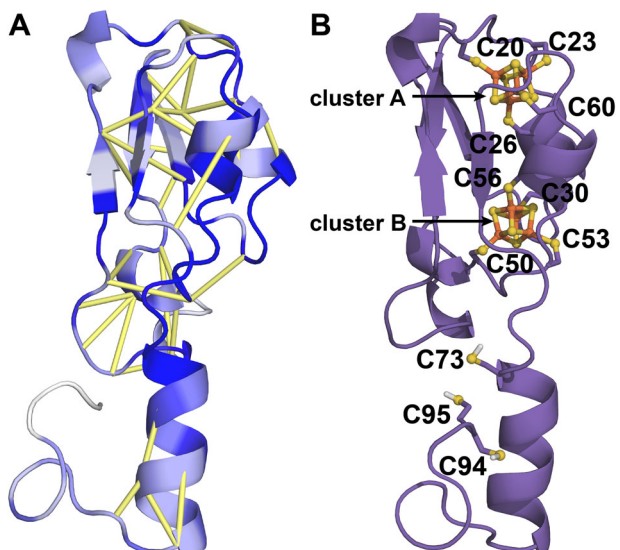

**Fig. 5 Structural Model of HgcB. a** Model of HgcB with predicted contacts shown as yellow bars and residues colored according to sequence conservation (dark blue = highest). **b** Location of conserved Cys residues and incorporation of [4Fe-4S] clusters.

Arg250 (N)–Asp8 (OD2), and Tyr303 (O)–Asp8 (N). The α-helical extension of HgcB interacts with TM helices 4 and 5 in HgcA, which protrude above the expected position of the membrane head groups. All contacts between the C-terminal extension and the TM helices of HgcA are nonpolar.

The distance between the closest Fe atom in cluster B and Co in the assembled model is 14.9 Å. The strictly conserved Cys73 in HgcB is located at the beginning of the C-terminal extension and is oriented away from the corrinoid in the CBD (Fig. 6). The C-terminal cysteines in HgcB (Cys94 and Cys95) are located at the end of a long, disordered tail, which is likely to be highly flexible. Both Cys73 and the $B_{12}$ cofactor are accessible by Cys94 and Cys95, suggesting a possible role of the cysteine pair in the transfer of $Hg^{2+}$, $[CH_3Hg]^+$, or both.

**Oligomerization state.** Several pieces of evidence suggested that HgcAB could function as a dimer of heterodimers, i.e., $(HgcAB)_2$:

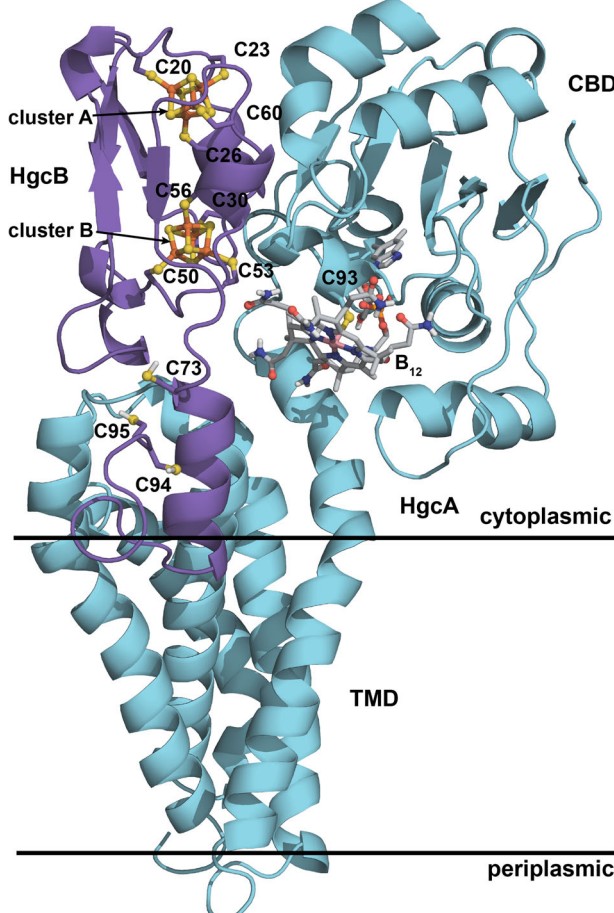

**Fig. 6 Model of the HgcAB complex.** Key features are labeled.

(i) Helices 1 and 5 in the TMD are not predicted to contact each other (Fig. 4), which suggests that the TMD may not form a tight, cylindrical bundle but may instead be more open or splayed out and may interact with another protein. (ii) The closest structural homolog to the TMD model identified by the Dali server is a homodimeric $Mg^{2+}$ transporter (PDB entry 2YVX)[30]. (iii) There appears to be self-complementarity in the shape of the HgcAB subunit, particularly in the TMD. (iv) Three functionally important residues in HgcB, Cys73, Cys94 and Cys95 are all oriented away from the $B_{12}$ cofactor in the HgcAB model (Fig. 6), but these residues in one HgcB protomer would be oriented toward the corrinoid in the opposite HgcA protomer in a dimer of heterodimers model. (v) Some of the predicted contacts, particularly in the TMD, are relatively long in the model and could potentially be interpreted as inter-oligomeric contacts. We therefore explored this possibility by performing symmetric docking[34] of two copies of HgcAB using ambiguous restraints. However, we found that the inter-oligomeric contacts were all longer and therefore less favorable than those in the original HgcAB model (Supplementary Fig. 5). Thus, the present coevolution analysis appears to support a 1:1 rather than a 2:2 oligomerization state.

**Phylogenetic analysis.** In addition to providing input for coevolution analysis, the deep multiple sequence alignment obtained in this work enables an unprecedented phylogenetic analysis of HgcA diversity in nature. It has been shown previously that the phylogeny of HgcAB is not congruent with that of Bacteria and Archaea species, suggesting the genes have been horizontally

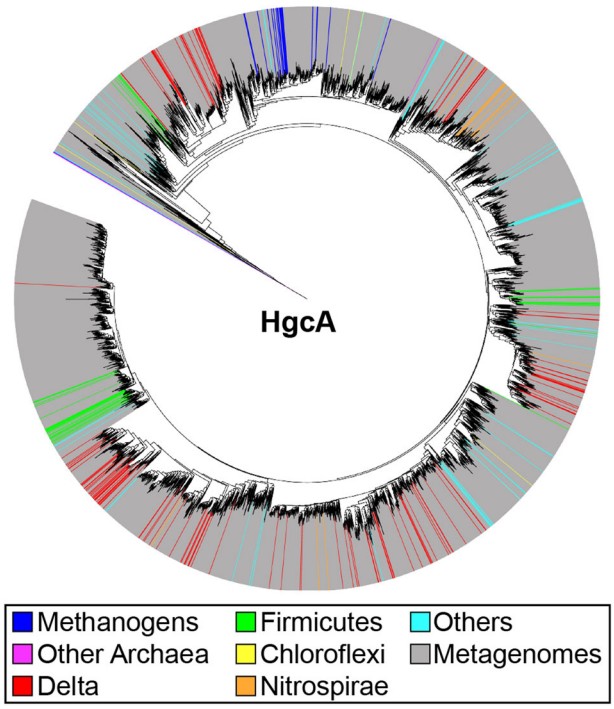

**Fig. 7 Phylogenetic tree of HgcA.** See "Methods" for details.

transferred across the different microbial lineages[5]. The more than tenfold expansion of the number of available sequences based on more recent metagenomes and additional cultured organisms provides much deeper insight into the diversity of Bacteria and Archaea that we predict to be able to methylate mercury, in a variety of environments. Although HgcA sequences from methanogens appear to remain confined to a single major clade, the genes from important methylating bacteria such as Deltaproteobacteria and Firmicutes are distributed across three or four distinct clades, suggesting multiple horizontal gene transfer events followed by independent diversification (Fig. 7 and Supplementary Data 3). The various groups of HgcA also include sequences from a variety of cultured bacterial phyla (including Chloroflexi, Nitrospirae, Spirochetes, Bacteroidetes), but also phyla with few or no cultured representatives (e.g., Raymondbacteria, Saganbacteria, Lentisphaerae). Several archaeal phyla with no cultured representative also appear to include potential methylators, such as Heimdallarchaeota and Theionarchaea. Interestingly, distinct sequence clades composed of dozens of metagenomic sequences cannot be assigned to any specific microbial taxa, suggesting we still have much to learn about the diversity of bacteria and archaea that can methylate mercury.

## Discussion

We have combined coevolution-based contact prediction and Rosetta modeling to generate a model of the HgcAB complex, which is responsible for Hg methylation in anaerobic microorganisms. This system is challenging to model because HgcA includes a transmembrane domain with no detectable sequence homology to any structurally characterized protein, and the complex consists of a unique heterodimeric structure in which the two domains of HgcA do not interact with each other but are instead bridged by interactions with HgcB. In addition, both proteins bind complex metal cofactors, which we have confirmed experimentally through heterologous expression and UV–visible spectroscopic characterization. These cofactors, vitamin $B_{12}$ and

two [4Fe-4S] clusters were incorporated into the model, which is consistent with available data from in vivo site-directed mutagenesis experiments targeting highly conserved residues in both HgcA and HgcB[10].

Some of the predicted residue-residue contacts in the fully assembled model are longer than expected (Supplementary Fig. 3), suggesting that structural rearrangements (i.e., domain motions) may occur during catalysis[35]. The closest Fe atom from [4Fe-4S] cluster B is ~15 Å from the Co center in the $B_{12}$ cofactor. However, it is likely that the CBD can move slightly closer to enable efficient electron transfer. Corrinoid-dependent enzymes with Rossmann domains often bind to $(\beta/\alpha)_8$ triosephosphate isomerase (TIM) barrel proteins to perform tightly controlled radical chemistry[36]. In addition, the CBD of the closest known homolog of HgcA, the corrinoid/iron–sulfur protein (CFeSP), is known to undergo large-scale conformational rearrangements, as revealed by X-ray co-crystal structures with its methyltransferase, a TIM barrel protein[8]. In the HgcAB model, the CBD is oriented toward the expected location of the membrane surface (Fig. 6). Such a conformation would preclude the approach and binding of a relatively large TIM barrel protein, suggesting that movement of the CBD would be required to accommodate a TIM barrel protein as a methyl donor.

The C-terminal tail of HgcB from *D. desulfuricans* ND132 includes a pair of cysteine residues (Cys94 and Cys95). Pairs of cysteines are commonly observed in proteins and enzymes involved in metal trafficking and detoxification, such as the proteins and enzymes encoded by the *mer* operon in Hg-resistant bacteria[37]. For example, the mercuric reductase (MerA), which catalyzes the reduction of $Hg^{II}$ to $Hg^0$, includes two Cys residues at its C-terminus that acquire $Hg^{II}$ and then transfer it to another pair of Cys residues in the active site. Whereas a double mutant of MerA in which both C-terminal Cys residues were substituted with Ala retained <0.1% of wild-type activity, a single Ala mutant maintained the same activity as the wild-type enzyme when an exogenous small-molecule thiol was present[38]. These findings suggest that when one of the Cys residues in the pair is replaced with Ala, a small-molecule thiolate can substitute for the missing Cys to satisfy the valence of $Hg^{II}$. However, loss of both Cys residues completely eliminates the tether that binds and properly positions $Hg^{II}$, resulting in a major reduction in activity.

Formation of MeHg by HgcAB has been previously proposed to proceed through a multi-step reaction involving (i) reduction of the corrinoid cofactor to form a $Co^I$ species, (ii) methylation of the $Co^I$ center to form a $CH_3$-$Co^{III}$ species, and (iii) methyl transfer to a $Hg^{II}$ substrate to form $[CH_3Hg^{II}]^+$ (Fig. 8a)[1]. The reduction step is presumed to be carried out by HgcB. The reduction potentials of the [4Fe-4S] clusters in HgcB and the corrinoid bound to HgcA have not been reported. However, parallels to CFeSP, in which a single [4Fe-4S] cluster serves a reductive activation role[39,40], would put the $Co^{II/I}$ couple below −500 mV versus SHE. Loss of the axial Cys93 ligand is expected upon reduction to $Co^I$ to give a four-coordinate complex, which is supported by density functional theory (DFT) calculations[11]. Subsequent oxidative addition of the methyl group and coordination of Cys93 from HgcA by the reduced corrinoid form the proposed active species for mercury methylation. The Hg substrate that is then methylated by HgcA to produce methylmercury is not known, but is assumed to be a $Hg^{II}$ bis(thiolato) species.

Our model provides insight into how HgcAB orchestrates the transfer and transformation of Hg. Specifically, we propose that Cys94 and Cys95 from HgcB acquire $Hg^{II}$ (from an unknown source) and deliver it to the corrinoid cofactor for methylation (Fig. 8b). The Hg methylation step has been proposed to proceed through either a methyl anion transfer or radical ligand exchange pathway[1,11]. A relativistic DFT study found that the latter

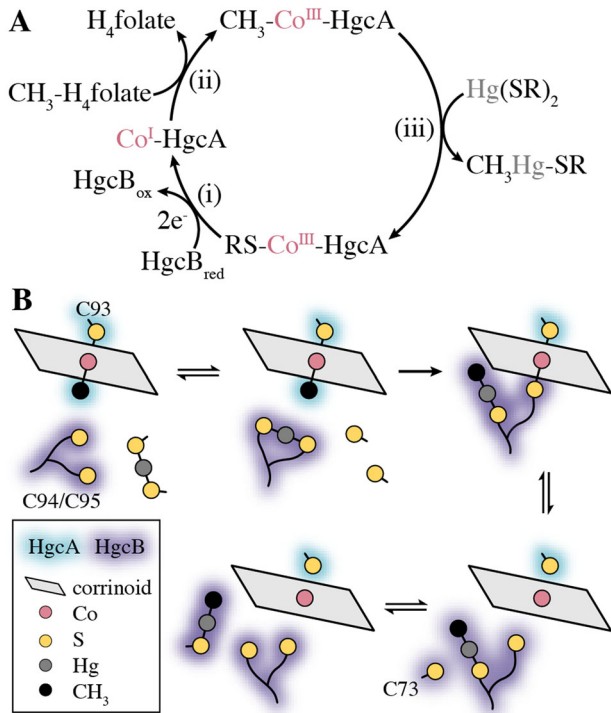

**Fig. 8 Mechanism of HgcAB. a** Proposed Hg methylation cycle from refs. [1,11,41]. **b** Proposed pathway for Hg[II] acquisition, methylation, and [CH$_3$Hg]$^+$ release.

pathway is energetically more favorable when spin-orbit effects were taken into account[41]. Assuming that the reaction proceeds through radical ligand exchange, a crosslinked HgcB-Cys94/95(Sγ)–Co[III]–(Sγ)Cys93-HgcA intermediate would be formed. Reduction of the Co center to Co[I] would then release both thiolate ligands and allow the C-terminal tail to deliver [CH$_3$Hg]$^+$ to Cys73 from HgcB. Either of the C-terminal cysteines (Cys94/95) could facilitate delivery of the [CH$_3$Hg]$^+$ product, as only a single Cys thiolate is required to bind this species. An exogenous thiolate, possibly a cysteine residue on a protein, would then displace Cys73 to liberate [CH$_3$Hg]$^+$ from HgcB, completing the reaction cycle. We expect that this structural model of HgcAB will facilitate the development of hypotheses addressing more detailed structural and functional questions that can then be tested experimentally.

## Methods

**Heterologous expression of His-tagged HgcA.** Full-length HgcA was produced as an N-terminal His-tagged construct (His-HgcA) and was co-transformed into *E. coli* BL21(DE3) cells along with pBAD42-BtuCEDFB, which encodes a cobalamin uptake operon[42]. His-HgcA was lysed and purified under anoxic conditions (<1 ppm oxygen) using Ni-NTA resin (Qiagen).

**Corrinoid extraction and UV–visible characterization.** Purified HgcA, which dissolved in Na phosphate buffer pH 7.4, was incubated with KCN (91 mM, final) and NaOH (23 μM, final) and heat-treated at 95 °C for 20 min. The solution was then centrifuged at 13,300 × *g* for 15 min, and the UV–visible spectrum of the supernatant was obtained with a Shimadzu UV-2600 spectrophotometer in a septum-sealed quartz cuvette versus a buffer-matched blank.

**Heterologous expression of MBP-tagged HgcB.** HgcB was produced as a maltose-binding protein fusion construct (MBP-HgcB) and co-expressed with the pRKISC vector, which encodes an inducible copy of the *E. coli isc* operon involved in iron–sulfur cluster assembly[17]. MBP-HgcB was lysed and purified under anoxic conditions (<1 ppm oxygen) using an amylose affinity resin. UV–visible spectra of MBP-HgcB were obtained with a Shimadzu UV-2600 spectrophotometer in a septum-sealed quartz cuvette, in a buffer containing 25 mM Na HEPES pH 7.5 and 2 mM dithiothreitol (DTT). The concentration of as-isolated (oxidized) [4Fe-4S]

clusters was estimated using the molar extinction coefficient of 4 mM$^{-1}$ Fe atoms at 390 nm[43].

**MSA generation and coevolution analysis.** The sequences of HgcA and HgcB from *D. desulfuricans* ND132[44] (UniProt IDs: F0JBF0 and F0JBF1, respectively) were selected for 3D structural modeling. In microbial genomes, *hgcB* is nearly always located immediately downstream of *hgcA*, which facilitated generation of the paired multiple sequence alignment. Initial alignments were generated by searching the UniProt20 database (2015_06) with *hhblits*[45] from HH-Suite[46] and then filtering the results with *hhfilter* to remove sequences with >90% identity and columns with >50% gaps. A hidden Markov model (HMM) was then generated from the alignment with *hmmbuild* from HMMER version 3.1b1 (http://hmmer.org) with default parameters, and *hmmsearch* was used to search a combined database consisting of JGI metagenomes (IMG/M)[24] and the UniRef100 database[23]. Filtering was performed to generate the final paired alignment. GREMLIN[47,48] was used to perform the coevolution analysis and predict intra- and interdomain contacts. A single GREMLIN calculation was performed on the paired multiple sequence alignment. The GREMLIN output provides predicted contacts that are ranked based on the strength of the coevolution signal between residue pairs. These raw contacts were then normalized and reweighted according to a previously described model that estimates the contact prediction accuracy from the normalized GREMLIN scores, the number of sequences in the MSA, and the length of the query sequence[22]. We also compared the contact map predicted by GREMLIN to the deep dilated residual network-based contact prediction server RaptorX-contact[49,50], which has been shown to be among the most accurate contact predictors currently available. Comparison of the contact maps from each server indicates that the two give similar results (Supplementary Fig. 6 and Supplementary Data 1). For consistency with previous work[25], we used the GREMLIN contacts to generate the model.

**HHΔ calculation.** *hhsearch* from HH-Suite was used to search the PDB70 database of hidden Markov models (HMMs) for homologous proteins with known structures using the HgcAB query HMM as input. For the resulting list of potential templates, HHΔ was calculated to determine if the multiple sequence alignment was closer to the query protein than a given structural homolog[47].

**Ab initio modeling.** The approach used to generate the model has been described previously[25]. Briefly, individual domains were folded with the standard Rosetta ab initio structure prediction method using restraints derived from the coevolution analysis. For each domain, we generated 10,000 models with sigmoidal restraints, 10,000 models with sigmoidal restraints and bounded restraints (with bounded restraints applied only during the centroid stage), and 4000 *map_align* models with sigmoidal and bounded restraints. The program *map_align*[25] identifies structural homologs by aligning contact maps predicted from coevolution analysis with contacts in experimentally determined structures, in this case a subset of the Protein Data Bank with a maximum of 30% mutual sequence identity[51].

The first nine residues of HgcA were excluded from the model because they are not highly conserved. The last ten residues of HgcB were not included in initial modeling but were added after the complex was assembled. Models were ranked by the sum of their Rosetta energy[52] and restraint score (scaled by a factor of 3). A diverse set of 30 top-scoring models selected on the basis of their pairwise TMscore[53] was then used as input for iterative hybridization[54]. The RosettaScripts interface[55] was used for both the *map_align* models and for iterative hybridization.

**Modeling of [4Fe-4S] clusters.** Consistent with the expected Cys coordination patterns from other dicluster ferredoxins, such as that from *Clostridium acidurici* (PDB entry 2FDN)[32], preliminary de novo models of HgcB with coevolution restraints suggested that one [4Fe-4S] cluster is bound to Cys20, Cys23, Cys26, and Cys60 and another is bound to Cys50, Cys53, Cys56, and Cys30. Thus, after a preliminary model of the HgcAB complex was generated, additional restraints were included in subsequent hybrid modeling to enforce geometries consistent with cluster binding. The C-terminal tail of HgcB was also introduced at this step. All Cys restraints were generated on the basis of the 0.94 Å resolution crystal structure of ferredoxin from *C. acidurici* (PDB entry 2FDN) and were the average values for the corresponding residues in each cluster. Harmonic distance restraints of 6.4 +/−0.5 Å were applied to all pairs of Sγ atoms among the four cysteines coordinated to each [4Fe-4S] cluster. Harmonic angle restraints were applied to Cα-Cβ-Sγ angles in each Cys residue as follows: Cys20 and Cys50, 114.6 +/−1 deg; Cys23 and Cys53, 116.9 +/−1 deg; Cys26 and Cys56, 112.9 +/−1 deg; Cys30 and Cys60, 108.9 +/− 1 deg. Circular harmonic restraints were applied to the C-Cα-Cβ-Sγ dihedrals in each Cys residue as follows: Cys20 and Cys50, 56.1 +/−2.3 deg; Cys23 and Cys53, −52.7+/−2.3 deg; Cys26 and Cys56, −71.6 +/− 2.3 deg; Cys30 and Cys60, 58.4 +/−2.3 deg. Explicit [4Fe-4S] clusters were placed into the final model by aligning the Sγ atoms of cluster-binding cysteines of the model with those in 2FDN.

**Modeling of the corrinoid cofactor.** The specific corrinoid cofactor used by HgcA differs from organism to organism. For example, the corrinoid used by most species of *Geobacter* is 5-hydroxybenzimidazolyl cobamide. However, the cofactor

used by ND132 is not known, so $B_{12}$ was used. The cofactor was first placed in the binding pocket by superposing the CBD onto an X-ray structure of CFeSP. Polar residues in the CBD of CFeSP that interact with the $B_{12}$ cofactor are conserved in HgcA. Thus, the following harmonic distance restraints were applied to facilitate cofactor binding in the HgcAB model: Thr60 (Oγ1)–$B_{12}$ (N3B), 2.9 +/− 0.1 Å; Thr66 (Oγ1)–$B_{12}$ (O4), 2.7 +/−0.2 Å; Val91 (N)–$B_{12}$ (O4), 3.0 +/− 0.05 Å; Ala153 (N)–$B_{12}$ (O6R), 3.1 + /−0.2 Å. Cys93 in HgcA was modeled as a chemically modified residue consisting of a coordinating bond between Sγ and the Co center in vitamin $B_{12}$ with a harmonic distance restraint of 2.5 + /− 0.1 Å and a Cβ-Sγ-Co harmonic angle restraint of 108 + /− 5 degrees. We then generated 1500 models with the Rosetta Relax application[56]. The model with the lowest Rosetta score was selected as the final model. The Dali web server[57] was used to identify structures in the PDB with folds that are similar to those of the HgcA and HgcB models. Figures were generated with PyMOL version 2.2.0[58].

**Phylogenetic analyses**. HgcA sequences identified in UniRef100 and IMG/M included 296 sequences from genomes of isolated bacteria and archaea and from taxonomically assigned uncultured organisms (assembled genomes from single cells or metagenomes), as well as ~4200 sequences (after filtering to a 90% identity cutoff) identified in bulk metagenomes. The sequences were aligned with Muscle (v. 3.8.425)[59] in Geneious (version 10)[60] and the alignment trimmed to eliminate highly variable positions (<30% overall similarity). A phylogenetic tree was constructed using FastTree (v. 2.1.12)[61] and visualized in iTOL[62].

**Statistics and reproducibility**. UV–visible spectra in Fig. 1 are single, representative spectra from multiple purifications of HgcA and HgcB, which were readily reproducible.

**Reporting summary**. Further information on research design is available in the Nature Research Reporting Summary linked to this article.

## Data availability

All data required to generate plots in Figs. 1 and 2 (Supplementary Data 1), the paired HgcAB multiple sequence alignment (Supplementary Data 2), HgcA multiple sequence alignment (Supplementary Data 3), and a complete list of metagenome datasets and associated references (Supplementary Data 4) are provided. Full gels and blots are shown in Supplementary Fig. 1. GREMLIN coevolution results including restraints used to generate the models can be obtained at https://gremlin2.bakerlab.org/preds.php?db=CASP12&id=278. The HgcAB model is deposited in the public Protein Data Bank (PDB) repository, PDB-Dev (https://pdb-dev.wwpdb.org)[63], under accession code PDBDEV_00000047. All other relevant data are available from the corresponding author upon request.

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

## Acknowledgements

This work was supported by the U.S. Department of Energy (DOE), Office of Science, Office of Biological and Environmental Research, through the Mercury Scientific Focus Area Program at Oak Ridge National Laboratory (ORNL) and the Laboratory Directed Research and Development program at ORNL, which is managed by UT Battelle, LLC, for DOE under contract DE-AC05–00OR22725. C.J.C. was supported by a National Science Foundation Graduate Research Fellowship under Grant No. 2017219379. S.W.R. was supported by NIH NIGMS grant R01GM124174. SO was supported by NIH grant DP5OD026389. G.A.P. and N.C.K. were supported by the U.S. DOE Joint Genome Institute, a DOE Office of Science User Facility, under contract no. DE-AC02-05CH11231 and used resources of the National Energy Research Scientific Computing Center, which is supported by the DOE Office of Science under contract no. DE-AC02-05CH11231. G.A.P. was also supported by the Hellenic Foundation for Research and Innovation (H.F.R.I) under the "First Call for H.F.R.I Research Projects to support faculty members and researchers and the procurement of high-cost research equipment grant", Grant ID: 1855-BOLOGNA. This research used resources at the Compute and Data Environment for Science (CADES) at ORNL. J.M.P. thanks J. Banfield for help with preliminary sequence searches.

## Author contributions

S.O., G.P., and N.C.K performed the metagenome searches; C.J.C., S.O., and J.M.P. performed the structural modeling; K.Z., K.W.R., and S.W.R. performed the cloning, expression, purification and spectroscopy; C.J.C., A.J., B.J.S., and J.M.P. performed the mechanistic analysis. M.P. performed the phylogenetic analysis; C.J.C., M.P., and J.M.P. prepared the manuscript with input from all other authors.

## Competing interests

The authors declare no competing interests.
