## [Peer Review File · Communications Biology]

Reviewers' comments:

Reviewer #1 (Remarks to the Author):

Comments to the Authors:

In the manuscript titled "Structure Determination of the HgcAB Complex Using Metagenome Sequence Data: Insights into Microbial Mercury Methylation" by Connor J. Cooper, Kaiyuan Zheng, Katherine W. Rush, Alexander Johs, Brian C. Sanders, Georgios A. Pavlopoulos, Nikos C. Kyrpides, Mircea Podar, Sergey Ovchinnikov, Stephen W. Ragsdale, and Jerry M. Parks, the authors modeled the HgcAB complex by combining metagenome sequence data, coevolution analysis, and ab initio structure calculations. The study further overexpressed HgcA and HgcB in *E. coli* and confirmed spectroscopically that they bind cobalamin and [4Fe-4S] clusters.

This study further claims that, the two domains of HgcA do not interact with each other but are instead bridged by interactions with HgcB, and provides insight into how HgcAB adjusts the transfer and transformation of Hg. Specifically, this paper propose that Cys9s4 and Cys95 from HgcB acquire HgII and deliver it to the corrinoid cofactor for methylation. Overall, this paper is well written and quite interesting. This study will help explore the microbial molecular Hg methylation mechanisms.

Minor changes: On the end of Result, check the sentence "the various groups of HgcAs also include 'sequences form' a variety of ..."

Overall Recommendation: Accepted with minor Revisions.

Reviewer #2 (Remarks to the Author):

This is a remarkably thorough manuscript. I am no structural biologist and I cannot comment on the experimental part.

Instead, I found the bioinformatics work extremely elegant, comprehensive and overall robust and convincing.

I have just two major comments on the coevolutionary part:

1) They used GREMLIN for contact prediction. How would it compare with other methods, such as direct Boltzmann learning? The proteins are small enough that it should be doable (although, given the present dire times, I am not necessarily insisting that the authors perform it if they can't access suitable computational infrastructures to do it, or find the time while confined at home)

2) How many coevolutionary predictions do they include in the Rosetta protocol? Are they weighted? I think that providing at least some hints of that in the Methods section could give the reader a feeling that it is not all a "black box".

I have no more comments, and once these have been addressed, I will clearly be in favour of publication, at least as far as the bioinformatics part is concerned.

Reviewer 1

In the manuscript titled “Structure Determination of the HgcAB Complex Using Metagenome Sequence Data: Insights into Microbial Mercury Methylation” by Connor J. Cooper, Kaiyuan Zheng, Katherine W. Rush, Alexander Johs, Brian C. Sanders, Georgios A. Pavlopoulos, Nikos C. Kyrpides, Mircea Podar, Sergey Ovchinnikov, Stephen W. Ragsdale, and Jerry M. Parks, the authors modeled the HgcAB complex by combining metagenome sequence data, coevolution analysis, and ab initio structure calculations. The study further overexpressed HgcA and HgcB in E. coli and confirmed spectroscopically that they bind cobalamin and [4Fe-4S] clusters.

This study further claims that, the two domains of HgcA do not interact with each other but are instead bridged by interactions with HgcB, and provides insight into how HgcAB adjusts the transfer and transformation of Hg. Specifically, this paper propose that Cys94 and Cys95 from HgcB acquire HgII and deliver it to the corrinoid cofactor for methylation. Overall, this paper is well written and quite interesting. This study will help explore the microbial molecular Hg methylation mechanisms.

Minor changes: On the end of Result, check the sentence “the various groups of HgcAs also include ‘sequences form’ a variety of ...”

Overall Recommendation: Accepted with minor Revisions.

Response: We have corrected this typo.

Reviewer 2

This is a remarkably thorough manuscript. I am no structural biologist and I cannot comment on the experimental part. Instead, I found the bioinformatics work extremely elegant, comprehensive and overall robust and convincing.

I have just two major comments on the coevolutionary part:

1) They used GREMLIN for contact prediction. How would it compare with other methods, such as direct Boltzmann learning? The proteins are small enough that it should be doable (although, given the present dire times, I am not necessarily insisting that the authors perform it if they can't access suitable computational infrastructures to do it, of find the time while confined at home).

Response: Our strategy was to apply the thoroughly validated protocol developed by members of the Baker lab, one of whom is a coauthor of our paper, to the HgcAB system (Ovchinnikov, *Science*, **2017**, 355, 294-298). This approach relies on contact predictions from GREMLIN and does not use neural networks, which we assume is what was meant by direct Boltzmann learning. Instead, our approach relies on Monte Carlo sampling with the Rosetta energy function to correct

any inaccurate contact predictions from GREMLIN. However, the field of protein structure prediction is evolving rapidly and advanced inter-residue contact (and interatomic distance) prediction algorithms based on convolutional neural networks have recently become the state of the art. Most currently used neural network-based approaches use GREMLIN (or its parallel implementation CCMpred) as an “ingredient” to their networks. Based on the Reviewer’s suggestion, we compared the contact map predicted by GREMLIN to the deep dilated residual network-based contact prediction server Raptor-X_contact (Xu J., *PNAS*, 2019, 116, 16856-16865; Wang S, Sun S, Li Z, Zhang R and Xu J. *PLoS Comput Biol*, 2017, 13, e1005324), which has been shown to be among the most accurate contact predictors available. Comparison of the contact maps from each server indicates that the two give similar results (**See figure below**). Thus, we would not expect major changes to the model if Raptor-X_contact or another accurate contact predictor were used. However, in future work we do intend to incorporate contacts and distances derived from deep learning into our modeling protocol.

Figure 1. Predicted contact maps for HgcAB predicted by GREMLIN (*left*) and Raptor-X_contact (*right*). A contact probability threshold of 0.5 was used in both cases. Darker shades of blue indicate higher probability.

We have revised the text (Methods, *MSA generation and coevolution analysis*, page 13) as follows:

We also compared the contact map predicted by GREMLIN to the deep dilated residual network-based contact prediction server Raptor-X_contact (Xu J., *PNAS*, 2019, 116, 16856-16865; Wang S, Sun S, Li Z, Zhang R and Xu J. *PLoS Comput Biol*, 2017, 13, e1005324), which has been shown to be among the most accurate contact predictors available. Comparison of the contact maps from each server indicates that the two give similar results (**Figure S#**). For consistency with previous work, we used the GREMLIN contacts here.

2) *How many coevolutionary predictions do they include in the Rosetta protocol? Are they weighted? I think that providing at least some hints of that in the Methods section could give the reader a feeling that it is not all a "black box".*

Response: To address the Reviewer's questions, we have added the following text to the revised manuscript (Methods, *MSA generation and coevolution analysis*, page 13):

A single GREMLIN calculation was performed on the paired multiple sequence alignment. The GREMLIN output provides predicted contacts that are ranked based on the strength of the coevolution signal between residue pairs. These raw contacts were then normalized and reweighted according to a previously described model that estimates the contact prediction accuracy from the normalized GREMLIN scores, the number of sequences in the MSA, and the length of the query sequence (Ovchinnikov, eLIFE, 2015).

I have no more comments, and once these have been addressed, I will clearly be in favour of publication, at least as far as the bioinformatics part is concerned.

We hope you find our revised manuscript suitable for publication in *Communications Biology*.

REVIEWERS' COMMENTS:

Reviewer #2 (Remarks to the Author):

I find very honest for the authors to compare with a different, and state of the art, predictor, such as RaptorX. The result confirms their predictions with GREMLIN.

What I meant by Boltzmann learning was, actually, the idea of learning the global co-evolutionary interactions via a MonteCarlo method, rather than using pseudo-likelihood (which is behind GREMLIN).

Yet it's ok for me as they did. Anything that could strengthen their findings was good.

I have no further concerns.